# Natural salivary gland barrier curtails Zika virus transmission in Thai *Aedes aegypti*

Channarong Sartsanga[1¤], Kittitat Suksirisawat[1], Jutharat Pengon[1], Chatpong Pethrak[1], Atiporn Saeung[2], Nunya Chotiwan[3], Kwanchanok Uppakara[3], Saranya Thaiudomsup[1], Phonchanan Pakparnich[1], Kittipat Aupalee[2], Kritsana Taai[4], Rinyaporn Phengchat[1], Anna B. Crist[5], Louis Lambrechts[5], Natapong Jupatanakul[ID][1]*

1 National Center for Genetic Engineering and Biotechnology (BIOTEC), Pathum Thani, Thailand, 2 Parasitology and Entomology Research Cluster (PERC), Department of Parasitology, Faculty of Medicine, Chiang Mai University, Chiang Mai, Thailand, 3 Chakri Naruebodindra Medical Institute, Faculty of Medicine Ramathibodi Hospital, Mahidol University, Samut Prakan, Thailand, 4 Faculty of Veterinary Medicine, Western University, Kanchanaburi, Thailand, 5 Institut Pasteur, Université Paris Cité, CNRS UMR2000, Insect-Virus Interactions Unit, Paris, France

¤Current address: Excellence Center for Biodiversity of Peninsular Thailand, Faculty of Science, Prince of Songkla University, Songkhla, Thailand
* natapong.jup@biotec.or.th

## Abstract

### Background

Vector competence is a critical determinant of arbovirus transmission dynamics, yet most studies in Thailand have primarily examined *Aedes aegypti* susceptibility to infection and systemic viral dissemination rather than the mosquito's potential to transmit infectious virions. Given that transmission potential varies among mosquito populations, identifying population-specific transmission barriers is crucial for optimizing vector control strategies especially under budget and resource constrains.

### Methodology/principal findings

This study assessed the Zika virus (ZIKV) transmission potential of three *Ae. aegypti* populations from Thailand: two field-derived populations from Chiang Mai (CSP) and Nakhon Sawan (NAK) and a long-established laboratory strain (DMSC). Following oral exposure to a locally circulating ZIKV strain, viral transmission potential was evaluated. CSP mosquitoes demonstrated the earliest and highest ZIKV prevalence in salivary glands at 7 days post-infectious blood meal (dpibm), with DMSC and NAK populations reaching comparable infection levels at later time points. Despite this, NAK mosquitoes exhibited a strong barrier, resulting in significantly lower transmission potential. Genetic crosses revealed that this phenotype is inherited as an autosomal dominant trait and is similarly effective against dengue virus serotype 2 (DENV2), underscoring the broad-spectrum potential of NAK *Ae. aegypti* for transmission suppression. Furthermore, male NAK mosquitoes exhibited superior mating

**Data availability statement:** All relevant data are within the manuscript.

**Funding:** This work was supported by Thailand Program Management Unit for Human Resources & Institutional Development, Research and Innovation (PMU-B), NXPO (B17F640002 to NJ) and the National Research Council of Thailand (NRCT) : High Potential Research Team Grant Program (N42A650869 to NJ); the French Government's Investissement d'Avenir program Laboratoire d'Excellence Integrative Biology of Emerging Infectious Diseases (ANR-10-LABX-62-IBEID to LL) and MSDAVENIR (grant INTRANZIGEANT to LL); The National Science, Research and Innovation Fund NSRF via the Program Management Unit for Human Resources & Institutional Development, Research and Innovation (B13F670055 to CS). The personnel exchange for the collaboration was supported by the European Union's Horizon 2020 research and innovation program under the Marie Skłodowska-Curie grant agreement No 734486 (SAFE-Aqua). The funders had no role in study design, data collection and analysis, decision to publish, or preparation of the manuscript.

**Competing interests:** The authors have declared that no competing interests exist.

competitiveness, reinforcing their viability as a tool for vector control by population replacement.

## Conclusions/significance

Our findings highlight significant variability in ZIKV vector competence among Thai *Ae. aegypti* populations, emphasizing the importance of direct transmission assessments in vector competence studies. The discovery of a naturally occurring, heritable salivary gland escape barrier presents an opportunity for vector control strategies through NAK-based population replacement approaches. With their strong mating capability and broad arbovirus-blocking ability, NAK mosquitoes provide a natural alternative to *Wolbachia*-based and genetically modified mosquito interventions.

---

## Author summary

*Aedes aegypti* is the primary vector of arboviruses such as Zika virus (ZIKV) and dengue virus (DENV), but not all mosquitoes that become infected successfully transmit the virus. While most research in Thailand has focused on mosquito infection and virus dissemination, fewer studies have examined the ability of mosquitoes to transmit infectious virus through their saliva. We discovered a mosquito population from Nakhon Sawan (NAK) that exhibits a natural barrier to virus release from the salivary glands, leading to significant blocking of virus transmission in ZIKV-infected mosquitoes. This salivary gland escape barrier of NAK is a heritable autosomal dominant characteristic that also restricts DENV transmission. Given their high mating competitiveness, NAK mosquitoes present a promising candidate for population replacement strategies aimed at reducing arbovirus transmission. Our findings emphasize the importance of assessing transmission rates rather than relying solely on infection and dissemination rates. Furthermore, the discovery of a naturally occurring mosquito population with reduced transmission potential offers a practical and sustainable vector control approach. The implementation of NAK-based population replacement strategies could serve as an alternative to traditional control methods, such as insecticide application or genetically modified mosquito releases. Future research should focus on field trials to validate the efficacy of this approach in real-world settings and assess its long-term impact on virus transmission dynamics.

## 1. Introduction

Arthropod-borne viruses (arboviruses) are transmitted in the human population by arthropod vectors such as mosquitoes. These viruses, including Zika virus (ZIKV) and dengue virus (DENV), cause significant health and economic burdens worldwide [1,2]. A large body of evidence indicates that vector competence—the physiological ability of an arthropod to become infected with and subsequently transmit a

pathogen—varies among geographically distinct *Aedes aegypti* populations [3–6] and thus can influence virus transmission dynamics. Population replacement, a strategy that introduces mosquitoes with low vector competence to outcompete highly competent individuals, such as *Wolbachia*-infected mosquitoes, offers a promising alternative to conventional control methods [7]. This approach minimizes the ecological disruption associated with population suppression strategies and provides a sustainable means of reducing viral transmission rates [8]. However, the adoption of *Wolbachia*-infected mosquitoes for vector control in several countries including Thailand has been slow, hindered by regulatory, logistical, and public acceptance challenges.

ZIKV has been circulating endemically in Thailand since at least 2002, exhibiting a focal transmission pattern that contrasts with the widespread epidemics seen in other regions such as during the 2015–2016 epidemic in South America [9,10]. Although population immunity has been suggested as a factor driving down ZIKV cases, recent serological studies indicate that ZIKV seroprevalence remains low, particularly among individuals aged 21–30 years [10]. Thus, immunity alone does not fully explain the spatially restricted nature of ZIKV transmission. Other contributing factors, such as heterogeneous mosquito vector competence [11], may influence local outbreak dynamics and contribute to the observed focal transmission pattern.

During mosquito transmission, crossing the salivary gland escape barrier (SGEB) is a crucial last step for arboviruses to be released in the saliva during blood feeding. Previous studies have shown that the SGEB's influence on transmission efficiency depends on both vector and viral genetic factors [3,12–14]. However, the molecular mechanisms underlying virus escape into saliva remain incompletely understood [15]. Current evidence suggests that, following dissemination from the midgut, viruses circulating in the hemocoel must propagate in fat body or hemocytes before traverse the basal lamina surrounding the salivary glands through tracheoles or nerve tissues in order to infect acinar cells, which are organized around a central salivary duct [13,15,16]. Viruses are eventually released in the apical cavities where saliva is stored prior to its release into vertebrate hosts during probing and feeding [13].

In this study, we investigated the ZIKV transmission potential of different *Ae. aegypti* populations in Thailand and assessed the feasibility of leveraging naturally occurring low-transmission traits for population replacement strategies. We further explored the genetic basis and inheritance patterns of a naturally occurring SGEB—a critical bottleneck preventing viral release into mosquito saliva. By understanding this transmission-blocking trait, we aimed to inform future vector control approaches that utilize inherent biological barriers to limit arbovirus spread, offering a sustainable and non-insecticidal strategy for vector-borne disease control.

## 2. Materials and methods

### 2.1 Ethics statement

This study was carried out in strict accordance with the recommendations in the Guide for the Care and Use of Laboratory Animals of the National Institutes of Health. Mice were used only for mosquito rearing as a blood source, according to the protocol (BT-Animal 05/2564). Mosquito infection assays were performed following the approved protocol (BT-Animal 05/2564). Both animal protocols were approved by the BIOTEC Committee for the use and care of laboratory animals. Human erythrocytes were used for infection by artificial membrane feeding. Human blood was collected from adult volunteers in accordance with the approved protocol (NIRB-052–2563), and written informed consent was obtained from participants prior to collection.

### 2.2 Mosquito populations and maintenance

Three *Ae. aegypti* populations were used in this study: early passages (within 8 generations) of two field-collected populations from Nakhon Sawan (NAK) and Chiang Mai (CSP) and a long-established laboratory strain (DMSC). Field mosquitoes were collected as larvae and reared under standardized insectary conditions ($28\pm1°C$, 70% relative humidity, and a

12:12-hour light/dark cycle). The laboratory strain (DMSC) that has been maintained at BIOTEC for over 36 generations under identical rearing conditions and used in our previous ZIKV infection study [17] was also included in the infection experiments. Larvae were provided powdered fish food (Tetra Bits, Tetra GmbH, Germany), and adults were provided with a 10% sterile sucrose solution *ad libitum*. To obtain the eggs for colony maintenance, mosquitoes were allowed to feed on ICR mice anesthetized with 2% Avertin (2,2,2-Tribromoethanol, Sigma, T48402).

## 2.3 Virus strains and propagation

ZIKV strain SV0010/15 isolated from a human patient in Thailand in 2015 and dengue virus serotype 2 (DENV2) strain DKA-8521 isolated from a human in Malaysia in 2009 were used for infection experiments. The *Aedes albopictus* cell line C6/36 (ATCC CRL-1660) was used for amplification of virus stocks. C6/36 cells were maintained at 28 °C in Leibovitz's-15 (L-15) medium (Sigma, L4386) supplemented with 10% fetal bovine serum (FBS, PAN BIOTECH, P30-3031), 2% tryptose phosphate broth (Sigma, T9157), 1X non-essential amino acids (Gibco, 11140–050), 1X Pen/Strep (100 U/ml of penicillin and 100 µg/ml of streptomycin, Cytiva, SV30010). Virus stocks were prepared in C6/36 cells according to a previously published protocol [18]. After the cells were cultured to 80% confluency in T-75 flasks (Corning, 430641U), all supernatants were removed and replaced with the virus stocks at the multiplicity of infection (MOI) of 0.1 in 5 mL L-15 medium without supplements for 2 hours. After virus incubation, the supernatant was removed and replaced with 15 mL of 2% FBS L-15 medium then further incubated at 28 °C. The supernatant was collected at 6–7 days post inoculation then supplemented with FBS to a final concentration of 20% and stored at -80 °C until further use.

## 2.4 Infectious virus titration

Plaque assay was used to determine infectious virus titers in both the virus stock and mosquito samples following a previously published protocol [17,19]. Briefly, virus samples were 10-fold serially diluted in Minimal Essential Medium (MEM) supplemented with 2% fetal bovine serum (FBS, PAN BIOTECH, P30-3031), 2% tryptose phosphate broth (Sigma, T9157), 1X non-essential amino acids (Gibco, 11140–050), 1X Pen/Strep (100 U/ml of penicillin and 100 µg/ml of streptomycin, Cytiva, SV30010). Diluted virus (100 µL) was added to BHK-21 cells seeded in 24-well plate at 80% confluency. Inoculated plates were then gently rocked at room temperature for 15 minutes before further incubation at 37 °C, 5% $CO_2$ for 45 minutes. After incubation, 1 mL of overlay medium (1% methylcellulose (Sigma, M0512) in MEM supplemented with 2% FBS and 1X Pen/Strep) was added to each well then further incubated at 37 °C, 5% $CO_2$ for 4 days for ZIKV or 5 days for DENV2. The wells were then fixed and stained with 200 µL of 0.5% crystal violet (Sigma, C6158) in 1:1 methanol/acetone fixative for 1 hour at room temperature. Stained plates were then washed under running tap water and air dried before plaque counting.

## 2.5 Mosquito infection by artificial membrane feeding

Mosquitoes were orally challenged with ZIKV and DENV2 using the Hemotek artificial membrane feeding system according to a previously published protocol [17]. Seven-day-old female mosquitoes were deprived of sucrose overnight before being exposed to an infectious blood meal containing 40% washed human erythrocytes mixed with ZIKV or DENV2 at titers of approximately 6 log10 PFU/mL. After feeding, mosquitoes were knocked down in a refrigerator for 15 minutes. Fully engorged females were then sorted on ice and maintained in waxed paper cups where they were maintained in climate-controlled chambers (28 °C, 80% relative humidity) with access to a 10% sterile sucrose solution.

## 2.6 Salivary gland dissection and saliva collection

For salivary glands dissection, mosquitoes were knocked down by refrigeration at 4 °C for 15 minutes, surface-sterilized with 70% ethanol, followed by two washes with 1XPBS. Dissections were performed in a drop of 1X PBS, and salivary glands were washed then collected in 150 µL of MEM supplemented with 10% FBS and 1X Pen/Strep. Samples were

stored at -80 °C before titration. To prepare the samples, tissues were homogenized using 0.5 mm glass beads with a Bullet Blender Tissue Homogenizer (NextAdvance). The homogenates were then serially diluted 10-fold and titrated by plaque assay as mentioned above.

Mosquito saliva was collected following a previously published protocol [18]. Briefly, mosquitoes were knocked down using a cotton ball soaked with triethylamine (Sigma, 808352). Saliva was collected by inserting proboscis into a pipette tip containing 20 µL of MEM supplemented with 10% FBS and 1X Pen/Strep. After a 45-minute salivation period, the medium from the pipette tips was mixed with 180 µL of MEM supplemented with 2% FBS and 1X Pen/Strep and immediately titrated using a plaque assay or stored at -80 °C for quantification of viral RNA.

To measure saliva droplet size, the mosquito proboscis was inserted into a glass capillary tube filled with a 1:1 mixture of Halocarbon oil 27 (H8773) and Halocarbon oil 700 (Sigma, H8898) for 45 minutes, after which the saliva volume was measured under a stereomicroscope (Nikon, SMZ745T) with digital camera (Lane optical technologies, MC4KW-G1). The size of the droplet was measured using using Pixit Pro camera software (Version 2.8 64bit).

### 2.7 Quantification of viral RNA by quantitative real-time PCR

To quantify viral RNA in the saliva, 2 µL of saliva sample was used in total 10 ul qPCR reaction. The reaction contained 0.2 µM each of ZV-SV15_86–111_F forward primer (5'-CGAGAGTTTCTGGTCATGAAAAACCC-3'), ZV-SV15_216–198_R reverse primer (5'-AGCAGAAGTCCGGCTGGCA-3') and ZV-SV15 146–167 probe (5'FAM-ATGCTAAAACGCGGAGTA GCCC-3'BHQ1) targeting 5'UTR of ZIKV, and 1X of SuperScript III One-Step RT-PCR System with Platinum *Taq* DNA Polymerase (Invitrogen, 11732–020). The thermal cycling conditions were as follows: 50°C for 15 minutes; 95°C for 3 minutes; followed by 40 cycles of amplification at 95°C for 3 seconds then 60°C for 30 seconds in BioRad CFX96 Touch Real Time PCR Detection System. Quantification of viral RNA copy was conducted using the standard curves of in vitro transcribed ZIKV 5'UTR region from $10^1$-$10^{10}$ copies/µL.

### 2.8 Salivary glands immunofluorescence

ZIKV infection in the salivary gland was visualized by immunofluorescence staining following the previously described midgut staining protocol [17] with slight modifications. Briefly, dissected salivary glands were placed on positively charged microscope slides (EMS, 71873–01) within the Immunofluorescence Chamber (EMS, 70330) containing 400 µL of 1X PBS. Salivary glands were fixed with 4% paraformaldehyde in 1X PBS at room temperature for 2 hours and then permeabilized using 2% Triton X-100 in 1X PBS for 15 minutes. After permeabilization, salivary glands were incubated in 3% bovine serum albumin (BSA, Himedia, MB083-25G) and 0.1% Triton X-100 in 1X PBS for 1 hour. Next, salivary glands were incubated with 200 µL of anti-flavivirus envelope protein 4G2 primary antibody produced in-house at 4 °C overnight. On the next day, salivary glands were washed 3 times with 400 µL of 1X PBS followed by 200 µL of 1:400 secondary antibody anti-mouse IgG Alexa 488 (Invitrogen, A28175) and 1:2000 Hoechst33342 (Invitrogen, 62249) in 1X PBS for 3 hours at room temperature. After additional three washes with 1X PBS, salivary glands were counter-stain with 200 µL of 1:200 Alexa 555-conjugated Wheat Germ Agglutinin (WGA) (Invitrogen, W32464) in 1X PBS for 30 minutes followed by three 1XPBS washes. Salivary glands were mounted onto glass slides in Vectashield Plus (Vector Laboratories, H1900). Images were taken under laser scanning confocal microscopes (Carl Zeiss LSM900) using 10X objective lens (Objective EC Plan-Neofluar 10x/0.3 Ph1 M27) and 40X objective lens (LD Plan-Neofluar 40x/0.60 Corr). The z-section distance was set to 2.5 µm, with a total Z-stack thickness of 25–45 µm. Salivary gland image stacks were merged using the z-projection function with the Max Intensity setting in Fiji (ImageJ, version 2.9.0/1.53t) [20].

### 2.9 Generation of cross families

*Ae. aegypti* from the high transmission potential DMSC population were crossed with those from the lower transmission potential NAK population to determine heritability of the transmission potential phenotype. The eggs from the DMSC

and NAK populations were hatched, and the resulting larvae were reared until they reached the pupal stage. To ensure that only virgin adult mosquitoes were used for crossbreeding, female and male pupae were separated based on size (with females being larger than males) before transferring to a cage and let them emerge to adults. Fifty females and 50 males were used in each cross. Four crossing families were established as follows: 1) DMSC♀ × DMSC♂ (DF × DM); 2) DMSC♀ × NAK♂ (DF × NM); 3) NAK♀ × DMSC♂ (NF × DM); and 4) NAK♀ × NAK♂ (NF × NM). These crosses were performed to investigate the heritability of the low transmission potential phenotype and to assess the fecundity of the F0 females from each family.

## 2.10 Blood meal size estimation

The blood meal size of the blood engorged mosquitoes was estimated by quantifying the amount of heme as previously described [19]. Recently blood engorged mosquitoes (approximately 30 minutes after offering the blood meal) were collected in 100 µL sterile Milli-Q water and stored at -80 °C until further analysis. The blood meal used in each blood feeding experiment was also stored at -80 °C for use as a standard. To measure heme amount, the stored mosquito samples were thawed and homogenized with 0.5 mm glass beads using Bullet Blender Tissue Homogenizer (NextAdvance). Supernatant of the homogenized samples was then collected after centrifugation at 8,000 xg, 4 °C for 2 minutes. Amount of heme in the supernatant was then measured using Heme assay kit (Sigma, MAK316) following the manufacturer's protocol. Briefly, 50 µL of supernatant was added to 96-well plate containing 200 µL of heme assay reagent, incubated at room temperature for 5 minutes, and the absorbance was measured at 400 nm. Fifty microliters of Milli-Q water were used as a blank control. For the standard curve, six adult female mosquitoes were homogenized in 600 µL of Milli-Q water. The resulting homogenate was divided into six aliquots, each supplemented with 0, 1, 2, 3, 4, or 5 µL of blood meal. A 50 µL aliquot of each standard sample was then added to a 96-well plate containing 200 µL of heme assay reagent. The absorbance was measured, and a linear standard curve was generated. Blood meal size for each mosquito was calculated by interpolating the absorbance values from the standard curve.

## 2.11 Generation of white-eye *Aedes aegypti* with CRISPR-Cas9-mediated knock out of *white* gene

A white-eye *Ae. aegypti* was generated with DMSC background strain using CRISPR-Cas9-mediated disruption of *white* gene (AAEL016999), an eye pigment transporter, by insertion this gene with white attP-ssODN. In brief, the mosquito embryos were injected with mixture of the 400 ng/µl EnGen Spy Cas9 NLS (New England Biolabs, M0646), 80 ng/µl each of 3 single-guide RNAs (sgRNAs) targeting exon 2 of *white* gene (5'-AAATGCGCAGACACAATCAT-3', 5'-GAC ACAATCATTGGAGCGCC-3', 5'-GTCGAA TGAAAGGCCTGTC-3'), and 200 ng/µl of single-strand donor containing white gene homology arm flanking attP site (5'-CTTCAGGAGCTTTCCTTAGCGAAATGCGCAGACA CGATCATTGGAG CGCCGTGCCCCAACTGGGGTAACCTTTGAGTTCTCTCAGTTGGGGGAGGTCGAATGAAAGGCCTGTCCGGAGG TGAACGGAAACGGTTGGCCTTTG-3'). Out of 200 injected embryos, 53 larvae survived and only one showed mosaic male white eyes. The mosaic male white-eye mosquito (generation zero, G0) was then crossed with wild-type DMSC to generate G1 offspring. Because *white* gene must be disrupted from both chromosomes for the white-eye phenotype, the G1 offspring were then inbred to obtain homozygous *white* gene knock out *Ae. aegypti*. After inbreeding, 254 white-eye G2 male offspring were obtained. They were then individually screened for the insertion of attP site using white HRL screening forward primer (5'-CATTCCCGCGCTTACAACTC-3') and white HRL screening R primer (5'-AACCCGGAAG TGGGTTCATC-3'). Out of 11 male mosquitoes with positive PCR result demonstrating insertion of white attP-ssODN, only one male mosquito showed correct sequence of attP inserted into the *white* gene. The validated male white-eye was backcrossed with the wild-type DMSC *Ae. aegypti* then inbred offspring for three additional generations to obtain homozygous female white-eye *Ae. aegypti*. The white-eye males and females were then mated to maintain the homozygous *white*⁻/⁻ line.

## 2.12 Mating competition assay

To evaluate mating competitiveness of male mosquitoes, a wild-type, black-eyed male (*white*+/+) from either the DMSC, CSP, or NAK population were allowed to compete with a white-eyed male (*white*−/−, DMSC background) for the opportunity to mate with a virgin, white-eyed female (*white*−/−, DMSC background). First, one black-eyed male and one white-eyed male were placed together in an oviposition tube with filter paper lining the side of the tube. Two to 3 hours later, a blood-fed white-eyed virgin female was introduced into the same tube, allowing the males to compete for copulation. Three days later, distilled water was added to the tubes to induce oviposition. All adult mosquitoes were removed 3 days later, and additional distilled water was added to fully submerge the eggs. One week later, the number of pupae was recorded and separated by eye color (black-eyed versus white-eyed) to calculate the percentage of offspring from each family.

## 2.13 Fecundity assay

Female fecundity was assessed by monitoring the number of eggs produced by F0 females in each crossing family. Three days after mating, female mosquitoes were allowed to feed on blood. Two to 3 hours later, blood-fed females from every family were individually transferred to an oviposition tube lined with filter paper. After 3 days, distilled water was added to the tubes to moisten the filter paper and facilitate oviposition. After an additional 3–4 days, females were removed, and the eggs were counted. Then, distilled water was added to promote hatching. Five days later, the number of larvae was recorded to calculate the hatch rate.

## 2.14 Statistical analyses

Statistical comparisons of infection prevalence, infectious titers of infected samples and proportion between white-eye and black-eye offspring between populations were conducted using estimation statistics [21] with the dabest package (version 2025.03.27) in python (version 3.12.11; https://acclab.github.io/DABEST-python/). Statistical comparisons of blood meal size and fecundity were conducted using Kruskal-Wallis followed by Dunn's posthoc test with the rstatix package (version 0.7.1) [22] in R (version 4.3.0). Graphs were generated using the ggpubr package (version 0.6.0) [23] in R.

## 3. Results

### 3.1 Variability in ZIKV transmission potential among Thai *Aedes aegypti* populations

First, we investigated whether populations of *Ae. aegypti* from Thailand exhibited variation in ZIKV transmission potential. To establish field colonies, immature stages of *Ae. aegypti* were collected from two provinces—Chiang Mai, which reported confirmed ZIKV cases, and Nakhon Sawan, which had no documented cases between 2016 and 2020. Both provinces are urbanized, with comparable population densities (~3,000 people/km²). We then evaluated ZIKV transmission potential in three *Ae. aegypti* populations: two field-collected populations from Chiang Mai (CSP) and Nakhon Sawan (NAK), and a laboratory strain (DMSC). Mosquitoes were orally infected with a locally circulating ZIKV strain from 2015, and salivary gland infection and transmission was assessed using plaque assays to quantify viral titers in the salivary glands and saliva over the course of infection.

Among the three populations, the CSP strain had the earliest detectable ZIKV in the salivary glands, with the significantly highest prevalence at 7 days post-infectious blood meal (dpibm) (87.5%), followed by the NAK (75%) and DMSC (54%) strains (Fig 1A). In addition to its earlier onset of infection, although not statistical significant, the CSP strain also displayed the highest initial viral titers in the infected salivary glands, while those from NAK strain had the lowest titers (Fig 1B). Over time, viral titers plateaued across all populations, resulting in similar salivary gland infection rates at later timepoints (14 and 21 dpibm).

While ZIKV was detected in the salivary glands at 7 dpibm, most saliva samples did not contain detectable levels of infectious ZIKV at this timepoint. Interestingly, at later timepoints, although CSP mosquitoes exhibited higher initial salivary

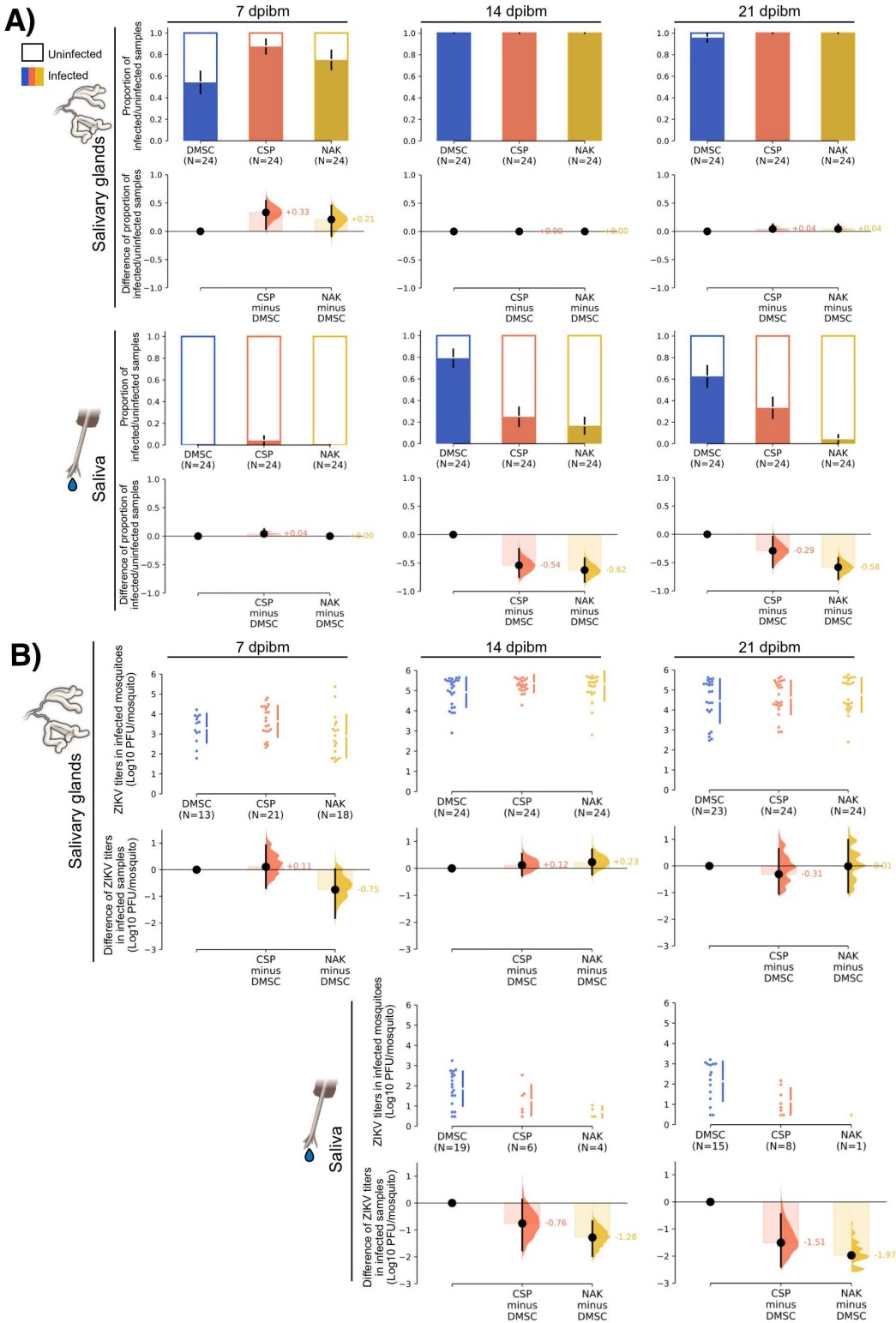

**Fig 1. ZIKV titers and prevalence in salivary glands and saliva of Thai *Ae. aegypti* populations.** ZIKV titers in salivary glands and saliva were measured at 7, 14, and 21 dpibm. Data were summarized from two blood feeding experiments with 12 mosquitoes from each. The measured blood meal titers were 5.5-5.6 log10 PFU/mL. (A) Estimation plots represent the infection prevalence (number of virus-positive samples/total number of samples). The dark color bars represent the proportion of virus-positive samples and the error bars represent a bootstrap 95% confidence interval (95% CI) from

5,000 resampling. Mean differences in infection prevalence relative to DMSC (effect size) are shown as distribution plots, with error bars indicating the 95% CI. (B) Swarm plots represent the distribution of viral titers in each virus-positive sample in log10-transformed PFU/mosquito. Median differences in virus titers relative to DMSC (effect size) are shown as distribution plots, with error bars indicating the 95% CI. Uninfected samples were not included in the analysis comparing virus titers. The images of the salivary glands and mosquito mouth parts were obtained from the NIH BioArt (bioart.niaid.nih.gov/bioart/363 and bioart.niaid.nih.gov/bioart/364).

gland infection rates, the DMSC strain had the highest ZIKV prevalence in saliva at both 14 and 21 dpibm, followed by CSP and NAK (Fig 1A). In addition to differences in the prevalence of saliva infection, the quantity of infectious virus in ZIKV-positive samples varied consistently among populations (Fig 1B), with DMSC having the highest while the NAK having the lowest transmission potential. The infectious titers in the ZIKV-positive saliva of CSP were 0.76–1.51 log10 PFU lower, whereas those of NAK were 1.28–1.97 log10 PFU lower than those of DMSC (Fig 1B, effect size). These findings indicate that while all tested *Ae. aegypti* populations were susceptible to ZIKV infection and allowed systemic viral dissemination, their ability to transmit the virus varied significantly, with NAK showing the least transmission potential.

### 3.2 The NAK salivary gland escape barrier does not reflect differences in cellular tropism, saliva production, or virus infectivity

We further investigated the nature of the SGEB responsible for the low transmission phenotype of the NAK population. One hypothesis was that the cellular tropism of ZIKV in salivary glands might vary between *Ae. aegypti* populations, with the saliva secretory cells of the NAK strain potentially being refractory to ZIKV infection. Such resistance could prevent the virus from being secreted into the salivary duct. To examine this hypothesis, we performed immunofluorescence staining at 13 dpibm to visualize ZIKV infection within the salivary gland tissue and the architecture of the saliva ducts. Each salivary gland of female *Aedes* mosquitoes consists of two lateral lobes and a medial lobe (Fig 2A). Each lobe contains a secretory epithelium surrounding a salivary duct, which serves as the outlet for saliva release. The complex composition of mosquito saliva is produced by secretory cells located in the proximal and distal regions of the lateral lobes, as well as the distal region of the medial lobe. Immunofluorescence analysis revealed that ZIKV was present and distributed throughout the salivary glands in both the proximal and distal regions of the lateral lobes across all three *Ae. aegypti* populations (Fig 2B). The proximal-lateral region exhibited the most concentrated fluorescent signal from antibody staining in all three *Ae. aegypti* strains, indicating a high viral load in this region. Given the significant differences in saliva virus prevalence and titers among the mosquito populations, if the cellular tropism explained the barrier, we would expect striking differences in infection patterns within the salivary gland tissue. However, immunofluorescent staining revealed no clear differences in infection tropism in the salivary glands of NAK mosquitoes. This suggests that the low transmission phenotype observed in the NAK population is unlikely a result from variations in cellular tropism within the salivary glands.

An alternative explanation for the differential transmission phenotype is variation in the quantity of saliva released by the mosquito populations. To test this, we measured the volume of saliva by allowing anesthetized mosquitoes to expel saliva into mineral oil and subsequently measuring the droplet size under a stereomicroscope. Our findings indicated that all mosquito populations produced similar amounts of saliva, regardless of their infection status (Fig 3). Thus, differences in saliva volume do not explain the observed variation in ZIKV transmission efficiency.

Finally, we investigated whether the low infectious virus titers in NAK saliva were due to the presence of antiviral factors or chemical properties that could inactivate the virus. If such inactivation were occurring, we would expect a difference between ratios of infectious and non-infectious virus among mosquito populations. To assess this, we performed linear regression analyses comparing infectious virus titers and ZIKV RNA copy numbers in the saliva of the three *Ae. aegypti* populations. The analyses revealed comparable slopes across all populations (Fig 4), indicating a consistent relationship between viral RNA levels and infectious titers. This suggests that the reduced infectious virus titers in NAK saliva are unlikely to result from inactivation of the virus.

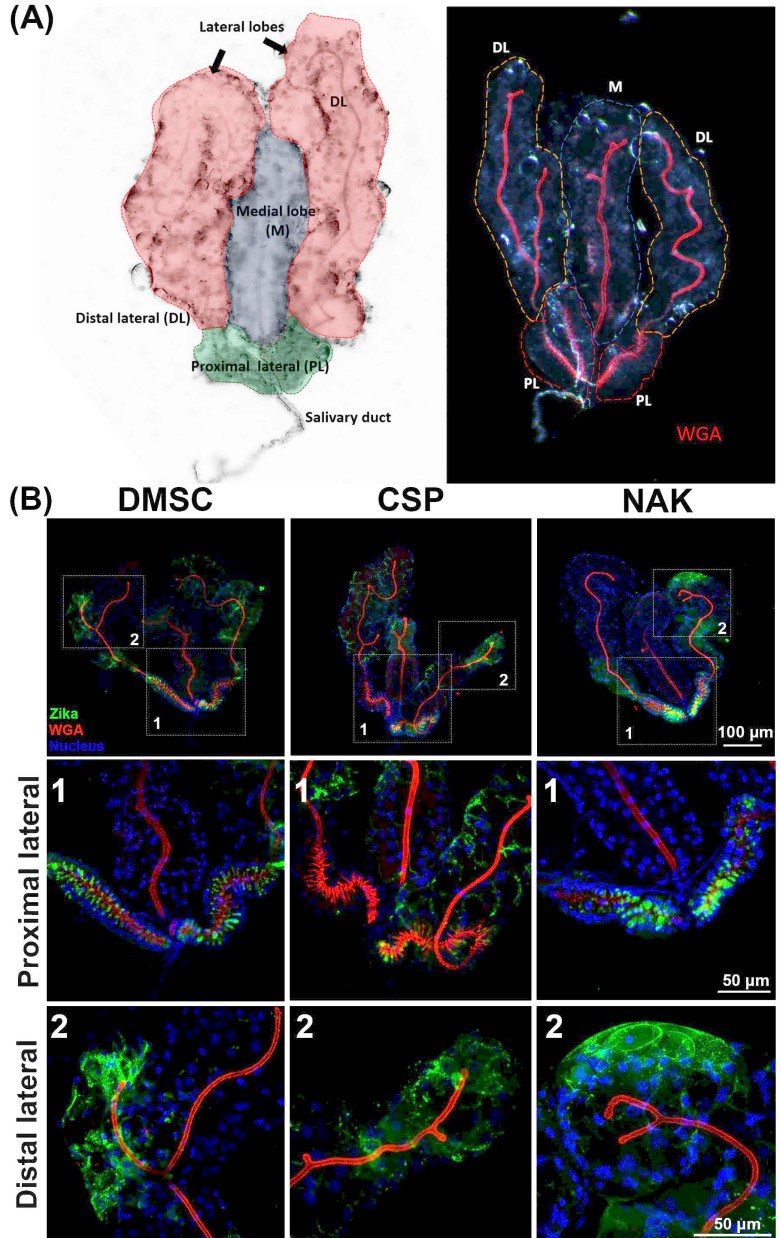

**Fig 2. Histological assessment of the ZIKV transmission barrier in NAK *Ae. aegypti*.** (A) Overall structure of *Ae. aegypti* salivary glands. Left: bright field image of salivary glands demonstrating regions of salivary glands. Right: Bright field image of salivary gland overlay with red channel demonstrating salivary duct stained with Alexa Fluor555-conjugated WGA. (B) Immunofluorescent staining of ZIKV in salivary glands of DMSC, CSP, and NAK *Ae. aegypti* at 13 dpibm. ZIKV infected cells were stained using the 4G2 primary antibody followed by a secondary Goat anti-mouse antibody conjugated with Alexa Fluor 488. Nuclei were stained with Hoechst-33342. Salivary ducts were stained using Alexa Fluor 555-conjugated wheat germ agglutinin. The top panel shows maximum projection of overall salivary gland images under a 4X objective lens. The zoomed images of proximal (1) and distal lateral (2) were maximum projected images of confocal z-stacks under a 20X objective lens.

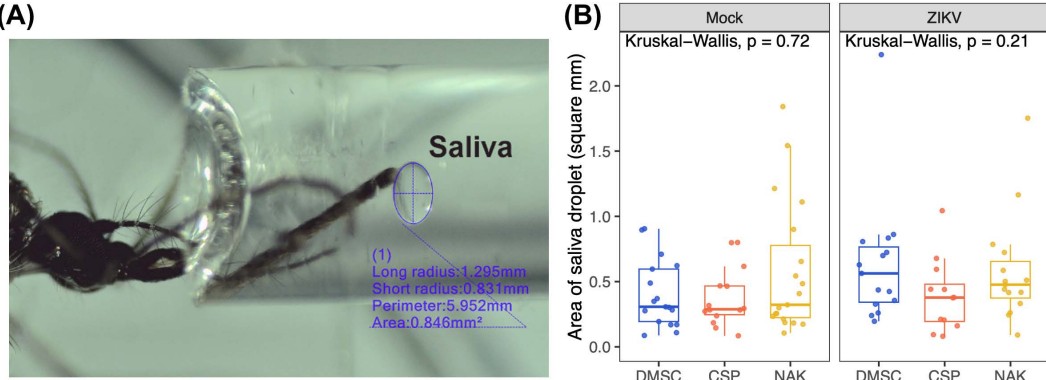

**Fig 3. Salivation assay comparing the amount of saliva excreted by each mosquito population.** Anesthetized mosquitoes were allowed to salivate in capillary tubes filled with an oil mixture of 50% Halocarbon oil 27 (H8773) and 50% Halocarbon oil 700 (H8898) for 30 minutes. Mosquitoes were infected by oral feeding with a blood meal containing 5.48-5.6 $\log_{10}$ PFU/mL. Blood meal containing MEM supplemented 10% FBS was used to feed the Mock group. Data were summarized from two salivation experiments. Box and scatter plots represent the distribution of the size of saliva excreted by individual mosquitoes as measured by the area of droplets (mm²). Statistical analysis was conducted using Kruskal-Wallis. Saliva image was captured by a stereomicroscope (Nikon, SMZ745T) with digital camera (Lane optical technologies, MC4KW-G1).

All these results collectively demonstrate that the ZIKV transmission barrier in NAK is not due to differences in cellular tropism, saliva production, or virus infectivity but rather an obstruction at the point of viral release from the salivary glands into saliva.

### 3.3 The SGEB of NAK *Ae. aegypti* is an autosomal dominant trait that blocks both ZIKV and DENV transmission

To explore the potential of utilizing the NAK *Ae. aegypti* population in a population replacement strategy to reduce arbovirus transmission, we investigated the genetic basis of the ZIKV SGEB. Specifically, we examined whether this trait is dominant, autosomal, and heritable by generating four reciprocal crosses between DMSC and NAK mosquito colonies: (1) DMSC♀ × DMSC♂ (DFxDM), (2) DMSC♀ × NAK♂ (DFxNM), (3) NAK♀ × DMSC♂ (NFxDM), and (4) NAK♀ × NAK♂. Salivary gland infection rates and transmission levels of these crosses were assessed at 14 dpibm.

We found that virus prevalence in the salivary glands was comparable in all crosses while titers in the salivary glands were comparable across three of the four crosses, except for NFxDM, which showed statistically significant increase in virus titers compared to the self-crosses (Fig 5). Despite similar salivary gland infection rates, ZIKV transmission potential in offspring from NAK hybrid crosses remained comparable to that of self-crossed NAK *Ae. aegypti* in both prevalence and titer levels (Fig 5). Additionally, the SGEB phenotype of NAK mosquitoes extended to DENV2 transmission, indicating a broader impact on arbovirus transmission (Fig 5). These findings indicate that the SGEB of NAK *Ae. aegypti* is an autosomal dominant trait, meaning that both male and female NAK mosquitoes could be used in population replacement strategies to reduce arbovirus transmission.

### 3.4 Mating competitiveness NAK *Ae. aegypti* in laboratory setting

To assess the feasibility of using NAK *Ae. aegypti* in a population replacement strategy, we evaluated the mating competitiveness of male NAK mosquitoes under laboratory conditions. A controlled mating competition experiment was conducted to quantitatively measure the mating competitiveness of male NAK *Ae. aegypti* relative to other mosquito populations using a transgenic white-eye (white gene knockout; *white*$^{-/-}$) *Ae. aegypti* strain generated in the DMSC background.

In this experimental setup, a single wild-type black-eye male (*white*$^{+/+}$) from either the NAK, CSP, or DMSC populations and a single male white-eye (DMSC background; *white*$^{-/-}$) were introduced into a 50 mL conical tube. The mosquitoes were

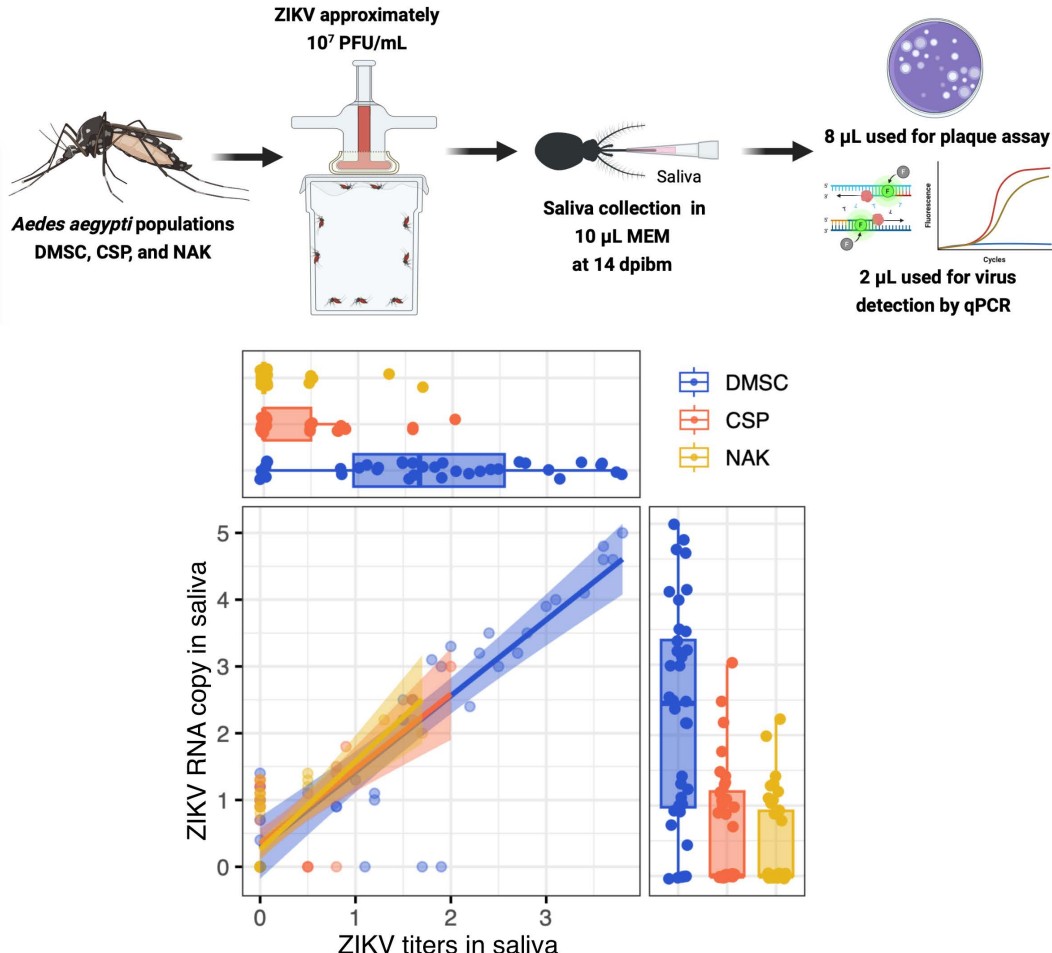

**Fig 4. Low transmission potential of NAK *Ae.aegypti* does not result from virus inactivation in saliva.** Scatter plot demonstrating relationship between ZIKV titers measured by plaque assay (x-axis) and ZIKV RNA copy measured by qPCR (y-axis) from each mosquito in $\log_{10}$ values. The horizontal and vertical box and scatter plots demonstrate projection of x-axis (titers) and y-axis (RNA copy) values from individual mosquito. Data were summarized from one blood feeding experiment. The measured blood meal titers was 7.2 $\log_{10}$ PFU/mL. Created in BioRender. Jupatanakul, N. (2025) https://BioRender.com/vu51gr2.

allowed a 3-hour acclimatization before the introduction of a blood-fed virgin female white-eye (*white*$^{-/-}$) DMSC. The mosquitoes were then maintained under controlled insectary conditions for 3 days prior to oviposition. Because male seminal fluid cause females to become unreceptive to a subsequent mating [24], all offspring from the same female inherited a single eye color phenotype in our experiment.

Our results showed that male NAK *Ae. aegypti* had the highest mating competitiveness, with 92% of the offspring families displaying black eyes, followed by DMSC and CSP males, with 71% and 30%, respectively (Fig 6). These findings demonstrate the superior mating competitiveness of male NAK *Ae. aegypti*, further supporting their potential utility in population replacement strategies for reducing arbovirus transmission.

### 3.5  NAK *Ae. aegypti* females exhibit high fecundity

To assess the potential of female NAK *Ae. aegypti* in population replacement strategies, we compared their fecundity to that of other *Ae. aegypti* strains. Our results demonstrated that female NAK mosquitoes laid the highest number of eggs

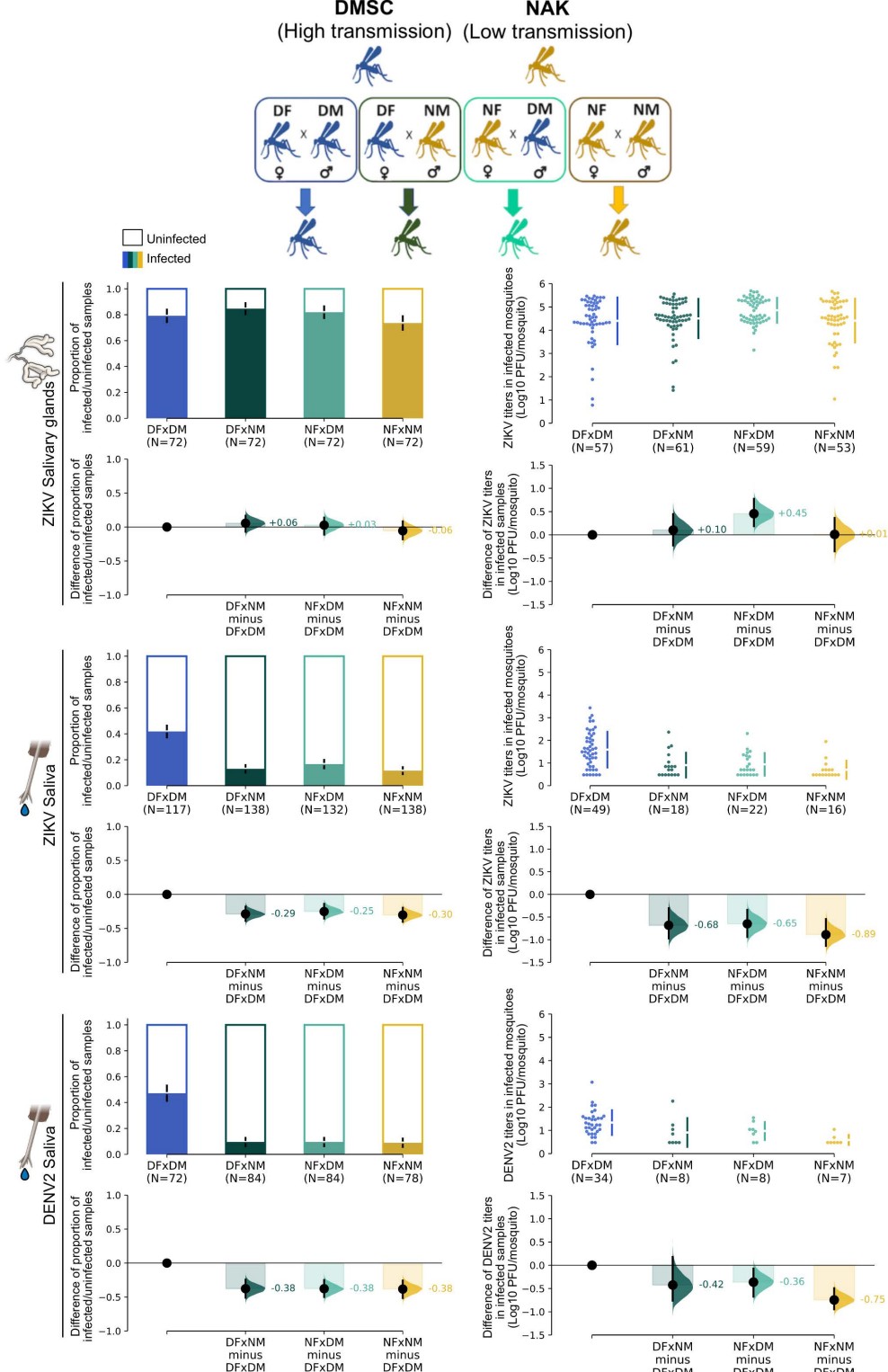

**Fig 5. Heritability of SGEB in the crosses between DMSC and NAK *Ae. aegypti*.** The offspring of each cross were fed on infectious blood meal containing 5.5-5.6 log10 PFU/mL of ZIKV or 5.9-6.1 log10 PFU/mL of DENV2. Virus titers in the salivary glands and saliva were measured at 14 dpibm.

The infection data were summarized from three blood feeding experiments except for ZIKV saliva that included an additional forth replicate. (Left) The dark color bars represent proportion of virus-positive samples and error bars represent a bootstrap 95% CI from 5,000 resampling. Mean differences in infection prevalence relative to DMSC (effect sizes) are shown as distribution plots, with error bars indicating the 95% CI. (Right) Swarm plots represent the distribution of viral titers in each virus-positive sample in log10-transformed PFU/mosquito. Mean differences in virus titers relative to DMSC (effect sizes) are shown as distribution plots, with error bars indicating the 95% CI. The images of the salivary glands and mosquito mouth parts were obtained from the NIH BioArt (bioart.niaid.nih.gov/bioart/363 and bioart.niaid.nih.gov/bioart/364).

compared to the DMSC and CSP females (Fig 7A). This higher reproductive output may be linked to their greater blood meal intake relative to the other populations (Fig 7B).

To determine whether this high fecundity was maintained during outbreeding, we crossed NAK females with DMSC males. We found that NAK females consistently produced a higher number of eggs, regardless of the paternal strain (Fig 7C). Furthermore, hatching rates remained comparable across all crosses (Fig 7C). These results indicate that NAK females retain their high reproductive capacity without any apparent disadvantages.

## 4. Discussion

Numerous studies have investigated the vector competence of *Ae. aegypti* mosquitoes, primarily focusing on their ability to become infected with arboviruses and develop disseminated infections, often extrapolating these findings to infer transmission potential [2,4,6,25,26]. In addition to laboratory-based infection studies, epidemiological models frequently

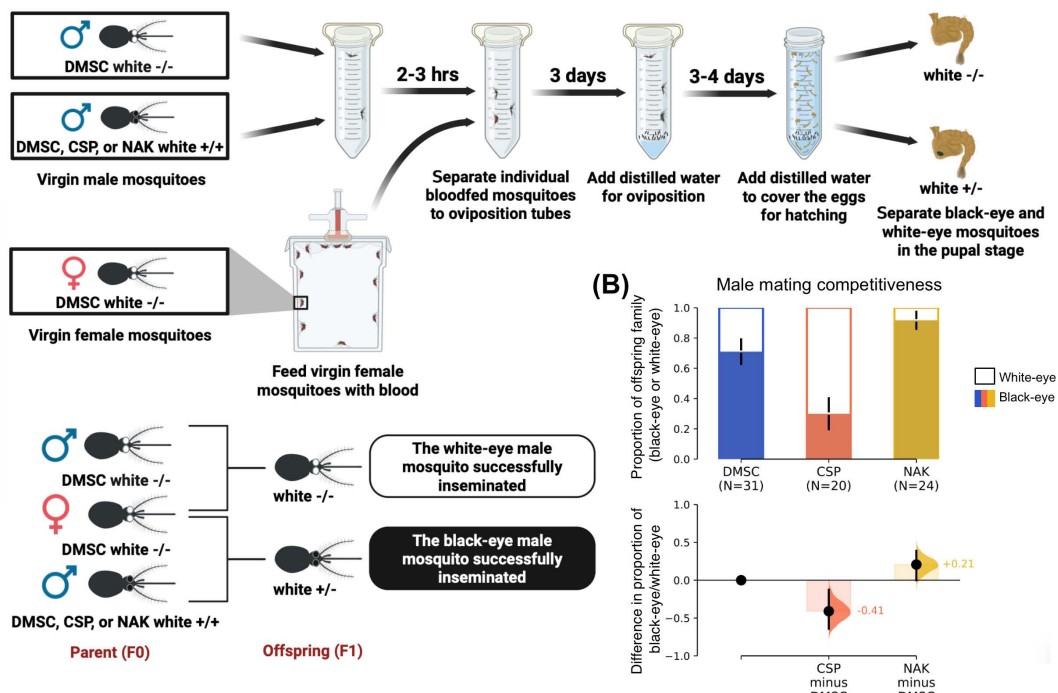

**Fig 6. NAK *Ae. aegypti* males exhibit high mating competitiveness.** (A) Schematic diagram of experimental design for mating competition assay between wild-type black-eye male (*white+/+*) from either the NAK, CSP, or DMSC populations and a single male white-eye (DMSC background; *white-/-*). (B) Bar chart illustrating the results of the mating competition assay. Statistical analysis comparing the eye color proportion between different population was conducted using estimation statistics. Each experiment was conducted with at least 10 mating trials from two independent rearing cohorts. Created in BioRender. Jupatanakul, N. (2025) https://BioRender.com/vu51gr2.

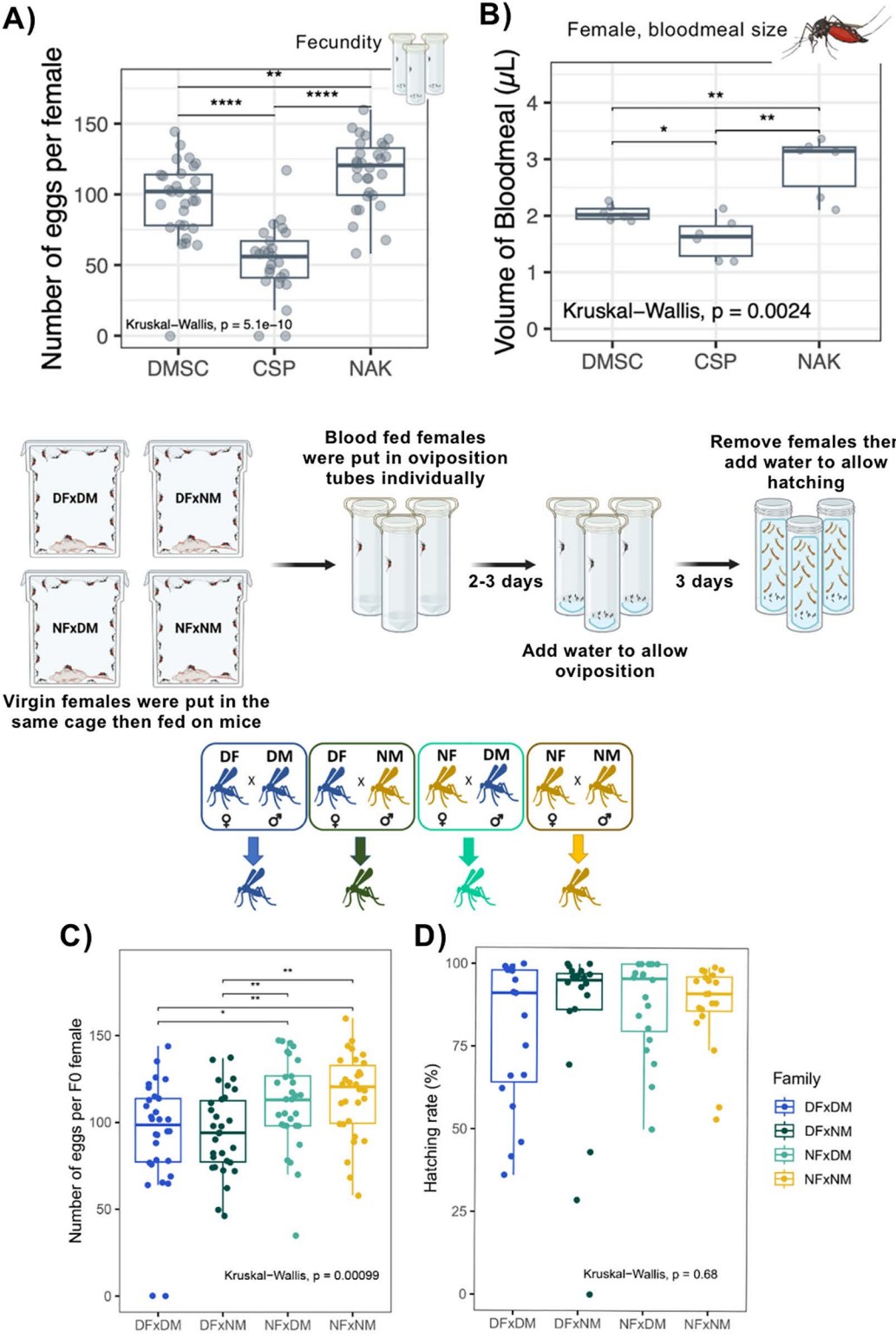

**Fig 7. NAK *Ae. aegypti* females exhibit high fecundity.** (A) Fecundity of each *Ae. aegypti* population when self-mated. Data were summarized from two blood feeding experiments with at least 10 mosquitoes from each. (B) Blood meal size of each *Ae. aegypti* population estimated by measuring heme content. Data were summarized from two blood feeding experiments with three mosquitoes from each. (C) Fecundity of F0 females from different

crosses represented by number of eggs laid per female mosquito. Data were pooled from three blood feeding experiments with at least 10 mosquitoes from each. (D) Hatching rate eggs from individual females. Data were pooled from two blood feeding experiments with at least 10 mosquitoes from each. Statistical analyses were conducted using Kruskal-Wallis followed by Dunn's post hoc test. *: p < 0.05, **: p < 0.01, ****: p < 0.0001. Created in BioRender. Jupatanakul, N. (2025) https://BioRender.com/vu51gr2.

incorporate mosquito infection status as a key parameter to estimate transmission risk [27]. For example, research on the spatiotemporal clustering of DENV transmission in Thai villages has linked human infections to the presence of infected mosquitoes. However, these studies rarely assess actual transmission potential—the ability of mosquitoes to release infectious virus into saliva. Several technical challenges have contributed to this research gap, including the difficulty of collecting live mosquitoes, the labor-intensive nature of transmission assays, and the complexity of direct virus quantification in saliva. Our findings reveal that disseminated infection or salivary gland infection does not always correlate with the presence of infectious virus in saliva, underscoring the limitations of using infection rates alone to infer transmission potential. The discrepancies between the salivary gland infection and escape barriers have previously been reported in different mosquito-virus models [13,28]. Furthermore, the impact of mosquito population variation in transmission potential on arbovirus epidemiology at a fine scale remains poorly understood. The threshold of transmission level required to sustain virus endemicity is unknown, representing a critical gap in our ability to predict outbreak dynamics and develop effective vector control strategies. Without empirical data on natural variation in transmission potential, intervention strategies may be misinformed, leading to inefficient resource allocation and suboptimal vector control efforts.

To address these gaps, a comprehensive epidemiological study is needed to investigate the relationship between arbovirus case numbers, mosquito density, seropositivity, entomological survey data, and vector transmission potential. Such an approach would provide a more precise understanding of how mosquito transmission efficiency influences disease dynamics at the population level. Our results demonstrated that recently colonized *Ae. aegypti* populations from Thailand exhibit varying levels of transmission potential, which may contribute to differences in virus endemicity across regions. Indeed, recent studies demonstrated that the relative proportion in genetic ancestry between two different forms of *Ae. aegypti – Ae. aegypti aegypti* (*Aaa*) and *Ae. aegypti formosus* (*Aaf*) – influences both vector competence and host feeding preference, thus shaping arbovirus transmission dynamics in Africa [11,29,30]. This underscores the need for further investigation into the role of vector population structure in driving transmission heterogeneity at finer geographic scales, which could inform targeted transmission intervention strategies. Thailand's robust dengue surveillance system, which includes weekly case reports down to the district level, provides a unique opportunity for spatiotemporal analyses of how population variation in vector transmission potential influences disease dynamics at a fine-scale. By integrating vector competence data with real-time epidemiological reports, future research could better elucidate the impact of mosquito population variability on arbovirus persistence and outbreak severity. Furthermore, recent advancements in detecting arboviruses in mosquito excreta and expectorates [31–33] offers new surveillance tools to evaluate transmission potential of wild mosquito populations in the endemic areas.

Vector control in Thailand continues to rely heavily on environmental management and insecticide-based interventions [34]. While innovative approaches such as the release of *Wolbachia*-infected mosquitoes have shown promise in reducing arbovirus transmission [35–38], policy makers have been reluctant to adopt these methods widely due to concerns about the introduction of non-native bacterial infections into mosquito populations. Our findings suggest that the high mating competitiveness of male NAK *Ae. aegypti* presents a viable, natural alternative for population replacement strategies. Since the observed SGEB is an autosomal dominant trait, introducing NAK males into wild populations could effectively propagate this characteristic, thereby reducing overall vectorial capacity. A male-only release strategy is particularly advantageous, as males do not bite, minimizing human nuisance, and their introduction does not significantly increase mosquito population densities as the population is still limited by the number of females in the environment. Additionally,

the contamination of the female mosquitoes during the release would be significantly less problematic compared to the use of *Wolbachia*-infected mosquitoes for population reduction.

While laboratory studies confirm high mating competitiveness, field validation is essential to assess performance under natural conditions, including dispersal ability and competition with wild-type males. Additionally, integrating this approach with spatial mapping of *Ae. aegypti* transmission hotspots could optimize deployment by targeting high-risk areas where mosquito populations exhibit high transmission efficiency. If successfully implemented, this strategy could provide a sustainable, non-chemical alternative to traditional vector control methods or the use of *Wolbachia*-infected or genetically modified mosquitoes. Importantly, it offers a practical and immediately deployable solution for reducing arbovirus transmission while public awareness, regulations, and policy frameworks continue to evolve to accommodate emerging next-generation technologies. Additionally, the implementation of male NAK releases could serve as a training platform for personnel, building the necessary technical capacity and infrastructure to support the future adoption of genetically modified mosquitoes and other advanced vector control technologies once they become regulatory and socially acceptable.

Another important takeaway from this study is that vectorial capacity and arbovirus infection kinetics in vectors is inherently dynamic, and evaluating vector competence at a single time point may provide an incomplete or misleading representation of a mosquito population's true ability to transmit a virus. Viral replication, dissemination, and transmission potential fluctuate over time [2,17,39], and a single time-point evaluation may either overestimate or underestimate vector competence depending on the stage of infection. For instance, in this study, salivary gland infection levels among Thai *Ae. aegypti* populations were significantly different at 7 dpibm; however, by 14 and 21 dpibm, infection levels plateaued, resulting in comparable salivary gland infection rates across all colonies. These findings emphasize the necessity of longitudinal assessments encompassing multiple time points to accurately characterize infection dynamics and transmission potential.

Although our current study successfully identified that the transmission barrier occurs during viral escape from the salivary glands, the nature and underlying mechanisms of this SGEB remain poorly understood. We demonstrated that the observed differences in transmission efficiency were not due to variations in tissue tropism, the quantity of saliva expelled by mosquitoes, or the presence of antiviral factors in the saliva. We also found that viral infection did not significantly alter mosquito salivation while a recent study reported reduced salivation following ZIKV and CHIKV infection [40]. The discrepancy may indicate that virus-induced behavioral changes may also be influenced by both host and viral genetic factors.

In addition to the potential mechanistic explanations that we addressed in this study there are several alternative mechanisms that deserve further investigation, including penetration through physical barrier, and cytopathic/apoptosis-induced virus release [15,41]. To further elucidate the nature of the SGEB in the NAK population, a comparative ultrastructural investigation of salivary glands across different mosquito populations is warranted to assess potential morphological or physiological differences that may contribute. Additionally, future comparative functional genomic and transcriptomic analyses will further provide genetic determinants for the SGEB, which when coupled with new CRISPR-based tools, can be used as a marker for the phenotype or further strengthen the barrier through genome/epigenome editing.

In mosquitoes, it has been demonstrated that anatomical barriers such as the midgut and salivary glands impose strong population bottlenecks resulting from stochastic reduction in population size and diversity, followed by population expansion and diversification [42–44]. Additionally, the vector genetic background has been shown to shape intra-host viral genetic diversity at both interspecific [44] and intraspecific [42] levels. Our panel of *Ae. aegypti* populations, which exhibit differing intra-host infection kinetics, provides a suitable model to investigate viral evolutionary dynamics under distinct selective pressures. This is particularly relevant during the final and arguably most critical step of the arbovirus transmission cycle—the escape of virus from the salivary glands into the saliva immediately prior to vertebrate host infection during blood feeding. The pronounced SGEB observed in the NAK population raises key questions about the nature of viral escape: Do specific viral subpopulations possess an enhanced capacity to traverse the SGEB, or is the process predominantly governed by stochastic mechanisms? Moreover, how does the strength of this barrier modulate the structure

and composition of viral populations at the point of transmission? Previous studies have reported varying degrees of convergent positive selection in virus populations in mosquito saliva during transmission across different vector-virus pairings [45–49]. While the mechanisms underlying the strength of such selection remain unclear, vector genotype is likely to play an important role in shaping these evolutionary outcomes. A combination of deep sequencing approaches to compare the composition of intra-host viral populations in salivary glands and saliva, coupled with transcriptomic or functional genomic analyses of the mosquito host simultaneously, could reveal the molecular basis of SGEB. These integrated approaches would not only clarify the mechanistic underpinnings of viral escape but also provide insight into the selective forces acting on virus populations at the point of transmission.

Taken together, the identification of a naturally occurring SGEB in a highly competitive male *Ae. aegypti* strain offers a promising, non-synthetic strategy for mitigating arbovirus transmission. This study represents an important step toward the development of more effective, evidence-based vector control programs in Thailand. Beyond its public health implications, our panel of *Ae. aegypti* populations with distinct infection kinetics provides a valuable model system for elucidating the molecular determinants of mosquito–arbovirus interactions, particularly during the critical transmission phase.

## Acknowledgments

The 4G2 antibody was a gift from Dr. Bunpote Siridechadilok (BIOTEC). The Thai ZIKV SV0010/15 was obtained from the Armed Forces Research Institute of Medical Sciences (AFRIMS) and the Department of Disease Control, Ministry of Public Health through the Cluster Program Management Office, NSTDA. The DENV2 DKA 8521 (Cat# NR-49749) was obtained through BEI Resources, NIAID, NIH, as part of the WRCEVA program. The NAK *Ae. aegypti* line was a gift from Professor Theeraphap Chareonviriyaphap, Department of Entomology, Faculty of Agriculture, Kasetsart University. We also thank Mr. Davide Milesi for his coding assistance with the data analyses.

## Author contributions

**Conceptualization:** Natapong Jupatanakul.

**Data curation:** Channarong Sartsanga, Natapong Jupatanakul.

**Formal analysis:** Channarong Sartsanga, Kittitat Suksirisawat, Chatpong Pethrak, Natapong Jupatanakul.

**Funding acquisition:** Louis Lambrechts, Natapong Jupatanakul.

**Investigation:** Channarong Sartsanga, Kittitat Suksirisawat, Jutharat Pengon, Chatpong Pethrak, Atiporn Saeung, Nunya Chotiwan, Kwanchanok Uppakara, Saranya Thaiudomsup, Phonchanan Pakparnich, Kittipat Aupalee, Kritsana Taai, Rinyaporn Phengchat, Natapong Jupatanakul.

**Methodology:** Channarong Sartsanga, Kittitat Suksirisawat, Jutharat Pengon, Chatpong Pethrak, Atiporn Saeung, Nunya Chotiwan, Kwanchanok Uppakara, Rinyaporn Phengchat, Anna B. Crist, Louis Lambrechts, Natapong Jupatanakul.

**Project administration:** Natapong Jupatanakul.

**Resources:** Atiporn Saeung, Nunya Chotiwan, Louis Lambrechts, Natapong Jupatanakul.

**Software:** Nunya Chotiwan, Kwanchanok Uppakara, Natapong Jupatanakul.

**Supervision:** Natapong Jupatanakul.

**Validation:** Channarong Sartsanga, Kittitat Suksirisawat, Chatpong Pethrak, Natapong Jupatanakul.

**Visualization:** Channarong Sartsanga, Kittitat Suksirisawat, Chatpong Pethrak, Natapong Jupatanakul.

**Writing – original draft:** Channarong Sartsanga, Natapong Jupatanakul.

**Writing – review & editing:** Channarong Sartsanga, Kittitat Suksirisawat, Jutharat Pengon, Chatpong Pethrak, Atiporn Saeung, Nunya Chotiwan, Louis Lambrechts, Natapong Jupatanakul.

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
