## [Decision Letter · Decision Letter 0]

25 Sep 2025

Response to Reviewers
Revised Manuscript with Track Changes
Manuscript

Shaden Kamhawi

co-Editor-in-Chief

Paul Brindley

co-Editor-in-Chief

**Journal Requirements:**

At this stage, the following Authors/Authors require contributions: Jutharat Pengon. Please ensure that the full contributions of each author are acknowledged in the "Add/Edit/Remove Authors" section of our submission form.

2) Some material included in your submission may be copyrighted. According to PLOSu2019s copyright policy, authors who use figures or other material (e.g., graphics, clipart, maps) from another author or copyright holder must demonstrate or obtain permission to publish this material under the Creative Commons Attribution 4.0 International (CC BY 4.0) License used by PLOS journals. Please closely review the details of PLOSu2019s copyright requirements here: PLOS Licenses and Copyright. If you need to request permissions from a copyright holder, you may use PLOS's Copyright Content Permission form.

Potential Copyright Issues:

- Please confirm (a) that you are the photographer of Figure 3A., or (b) provide written permission from the photographer to publish the photo(s) under our CC BY 4.0 license.

- Figures 4, 5, 6, and and 7.. Please confirm whether you drew the images / clip-art within the figure panels by hand. If you did not draw the images, please provide (a) a link to the source of the images or icons and their license / terms of use; or (b) written permission from the copyright holder to publish the images or icons under our CC BY 4.0 license. Alternatively, you may replace the images with open source alternatives. See these open source resources you may use to replace images / clip-art:

3) Please ensure that the funders and grant numbers match between the Financial Disclosure field and the Funding Information tab in your submission form. Note that the funders must be provided in the same order in both places as well.

**Reviewers' comments:**

**Key Review Criteria Required for Acceptance?**

**Methods:**

-Are the objectives of the study clearly articulated with a clear testable hypothesis stated?

-Is the study design appropriate to address the stated objectives?

-Is the population clearly described and appropriate for the hypothesis being tested?

-Is the sample size sufficient to ensure adequate power to address the hypothesis being tested?

-Were correct statistical analysis used to support conclusions?

-Are there concerns about ethical or regulatory requirements being met?

Reviewer #1: See attached comments.

Reviewer #2: Please see my general comments.

Reviewer #3: (No Response)

**Results:**

-Does the analysis presented match the analysis plan?

-Are the results clearly and completely presented?

-Are the figures (Tables, Images) of sufficient quality for clarity?

Reviewer #1: See attached comments.

Reviewer #2: Please see my general comments.

Reviewer #3: (No Response)

**Conclusions:**

-Are the conclusions supported by the data presented?

-Are the limitations of analysis clearly described?

-Do the authors discuss how these data can be helpful to advance our understanding of the topic under study?

-Is public health relevance addressed?

Reviewer #1: See attached comments.

Reviewer #2: Please see my general comments.

Reviewer #3: (No Response)

**Editorial and Data Presentation Modifications?**

Reviewer #1: (No Response)

Reviewer #2: Please see my general comments.

Reviewer #3: (No Response)

**Summary and General Comments:**

Reviewer #1: See attached comments.

Reviewer #2: General Comment: This study evaluated a salivary gland escape barrier in several populations of Ae. aegypti. The presence of this genetically inherited barrier to transmission may be incorporated into disease prevention strategies. I am glad to see that the authors used virus detection rather than RT-PCR to detect ZIKV. The mere detection of viral RNA does not mean that any infectious virus was present. Similarly, I am glad to see that the authors used “days post-infectious blood meal” rather than the more commonly and incorrectly used “days post infection.” If it was days post infection, then all of the mosquitoes would have been infected. Overall, this is a very informative study and involved a lot of well-designed work. Below are several suggested changes. Most are merely style/format corrections.

Specific Comments:

1. Lines 81-82: Why establish WNV, JEV, and YFV if they are not used again?

2. Line 86: As this is the first time that Aedes aegypti is mentioned in the body of the paper, shouldn’t it be written out?

3. Line 100: Should “This immunity alone…” be “Thus, immunity alone…?”

4. Line 114: Shouldn’t “human” be “humans” as it is referring to more than a single person. More importantly, as this is referring to SGEB in general, shouldn’t this be “vertebrate hosts…?”

5. Line 115: This should be in the past tense, so it should be “we investigated the…” Similarly, on line 117, “explore” should be “explored…” and on line 120. “we aim to…” should have been “we aimed to…”

6. Line 255: Should “the transmission potential NAK population” be “the lower transmission potential NAK population?”

7. Line 260: As “fifty” is not the first word in the sentence and is greater than 10, it should be “50 males…”

8. Lines 331-335: Unless they are the first word in a sentence, numbers of measurement should not be written out and numerals should be used, e.g., 3 hours or 3 days. Note, this was correctly done on line 362 and others.

9. Line 351: As ZIKV was established on line 80, there is no reason to reestablished here.

10. Lines372-377: Yes, looking at Figure 1, I can see that the ability of the NAK strain to transmit ZIKV was significantly lower than that of either of the other two populations. You might want to modify the sentence to, “…varied significantly with transmission rates for the NAK being significantly (p < 0.0001) lower than that for the DMSC (Figure 1).”

11. Line 474-475: Why reestablish dpibm as it was established on line 362.

12. Lines 489-490: While it is easy to calculate, viral titers to a hundredth of a log are usually not accurate and claim a degree of accuracy not supported by data. Therefore, I suggest that you change these to 5.5-5.6 log and 5.9-6.1 log…

13. Lines 590-591: Yes, the Aaa are more likely to feed on humans than the Aaf, and thus, are much more likely to be involved in the transmission of anthropomorphic viruses such as dengue and chikungunya. See the recent studies by Agha et al. Entomological assessment of dengue virus transmission risk in three urban areas of Kenya. PLoS Negl Trop Dis. 2019 Aug 23;13(8):e0007686 and Agha et al. Vector competence of populations of Aedes aegypti from three distinct cities in Kenya for chikungunya virus. PLoS Negl Trop Dis. 2017 Aug 18;11(8):e0005860. Not only is transmission efficiency critical, but feeding preference is also important

14. References: These need to be formatted properly.

a. Only the first word and proper nouns in a reference title should be capitalized. See references 5, 15, 24 and many others.

b. Genus and species names should be in italics. See references 2, 3 4, and others.

c. Why is the name of the editor of some of the PLoS journal papers included? See references 5, 12 and others.

d. Shouldn’t PLOS ONE or PLoS ONE be PLoS One? See references 5, 12, and others? You need to be consistent.

e. Why is the title of reference 28 in all caps?

Reviewer #3: To the Authors:

The manuscript entitled: “Natural salivary gland barrier curtails Zika virus transmission in Thai Aedes aegypti” by Sartsanga et al. is well written and clear. The study reports the discovery of a heritable salivary gland barrier, which may be used to form the basis of an alternative arbovirus control strategy using population replacement strategies as well as a tool to explore mosquito arbovirus interactions. Nice work.

PLOS authors have the option to publish the peer review history of their article (what does this mean? ). If published, this will include your full peer review and any attached files.

**Do you want your identity to be public for this peer review?** For information about this choice, including consent withdrawal, please see our Privacy Policy .

Reviewer #1: No

Reviewer #2: No

Reviewer #3: No

**Figure resubmission:**

**Reproducibility:** To enhance the reproducibility of your results, we recommend that authors of applicable studies deposit laboratory protocols in protocols.io, where a protocol can be assigned its own identifier (DOI) such that it can be cited independently in the future. Additionally, PLOS ONE offers an option to publish peer-reviewed clinical study protocols. Read more information on sharing protocols at https://plos.org/protocols?utm_medium=editorial-email&utm_source=authorletters&utm_campaign=protocols

---

## [Editor Report · Decision Letter 1]

20 Oct 2025

Dear Dr. Jupatanakul,

We are pleased to inform you that your manuscript 'Natural salivary gland barrier curtails Zika virus transmission in Thai Aedes aegypti' has been provisionally accepted for publication in PLOS Neglected Tropical Diseases.

Best regards,

Nikos Vasilakis

Section Editor

Álvaro Acosta-Serrano

Section Editor

Shaden Kamhawi

co-Editor-in-Chief

Paul Brindley

co-Editor-in-Chief

The revision has significantly improved the quality of the manuscript and critically addressed thoughtfully Reviewer's 1 concerns.

---

## [Editor Report · Acceptance letter]

Dear Dr. Jupatanakul,

We are delighted to inform you that your manuscript, "Natural salivary gland barrier curtails Zika virus transmission in Thai Aedes aegypti," has been formally accepted for publication in PLOS Neglected Tropical Diseases.

Best regards,

Shaden Kamhawi

co-Editor-in-Chief

Paul Brindley

co-Editor-in-Chief
